# Effects of Oral Probiotics on Subjective Halitosis, Oral Health, and Psychosocial Health of College Students: A Randomized, Double-Blind, Placebo-Controlled Study

**DOI:** 10.3390/ijerph18031143

**Published:** 2021-01-28

**Authors:** Dong-Suk Lee, Myoungsuk Kim, Seoul-Hee Nam, Mi-Sun Kang, Seung-Ah Lee

**Affiliations:** 1College of Nursing, Kangwon National University, Chuncheon 24341, Korea; ds1119@kangwon.ac.kr (D.-S.L.); ronjjang@naver.com (S.-A.L.); 2Department of Dental Hygiene, College of Health Science, Kangwon National University, Samcheok-si 25949, Korea; nshee@kangwon.ac.kr; 3R&D Center, OraPharm Inc., Seoul 04782, Korea; jieenkang@orapharm.com

**Keywords:** halitosis, probiotics, depression, self-esteem, oral health

## Abstract

Altogether, 81% of Korean college students experience halitosis and concomitant psychosocial problems such as depression and lowered self-esteem, as well as poor oral-health-related quality of life. Although halitosis causes many social and psychological problems among college students, there have been no reports of improvement interventions. This study aimed to identify the effects of ingesting tablets of the oral probiotic *Weissella cibaria* CMU (Chonnam Medical University, Gwangju, Korea) on halitosis and examine its effects on psychosocial indicators. This was a randomized, double-blind, placebo-controlled trial. The participants were randomly assigned to the experimental group or the control group. They ingested *W. cibaria* CMU or the placebo, depending on which group they belonged to, before going to bed daily for eight weeks. The measured indicators were subjective halitosis, subjective oral-health status, depression, self-esteem, and oral-health-related quality of life. Measurements were at baseline and eight weeks later. The participants showed statistically significant differences in subjective halitosis and oral-health-related quality of life. For college students with halitosis, intake of the oral probiotic for eight weeks could be a useful nursing intervention for reducing halitosis and improving oral-health-related quality of life.

## 1. Introduction

Halitosis is a condition in which an unpleasant smell caused by oral microbes comes out of a person’s mouth during exhalation, which can make others feel uncomfortable [1]. Halitosis is common occurrence; worldwide, prevalence of halitosis varied from approximately 5% to 65.9% [2,3]. It is also very common in Korean college students, with a prevalence of 81% [4].

Since someone with halitosis is concerned that other people will feel uncomfortable, he or she may experience mental stress; thus, it should be considered a social problem, in addition to being an individual problem. Halitosis is not only an indicator of poor oral health, but it also has an important influence on people’s psychosocial health [5]. Participants with subjective halitosis were found to be more anxious and to experience stress and depression more often [6]. Halitosis also negatively affects interpersonal relationships and quality of life, and can cause low self-esteem, depression, and suicidal intent [7].

The factors influencing the incidence of halitosis are diverse, including decreased salivation, fasting status, oral diseases, drug use, alcohol consumption, and smoking [8]. However, in most cases, the main contributing factor is bacterial adhesion on the teeth and tongue. More specifically, halitosis is mainly caused by bacterial by-products due to gram-negative anaerobic bacteria present in the tongue and their metabolites, like volatile sulfur compounds [9]. Volatile sulfur compounds (VSCs) are mainly composed of hydrogen sulfide (H_2_S), dimethyl sulfide [(CH_3_)_2_S], and methyl mercaptan (CH_3_SH), of which hydrogen sulfide and methyl mercaptan account for about 92% [10]. Suppressing oral bacterial growth has been shown to be most effective in suppressing halitosis [11]. Methods of reducing oral bacteria include using toothpaste, disinfectants, gargle solutions, tongue scrapers, and antibiotics. However, these methods can cause trauma to the tongue and eradicate not only harmful bacteria in the mouth, but also normal oral flora, which can destroy the balance of the microbial ecosystem in the oral cavity and thereby cause oral diseases [9]. Oral probiotics are known to be effective in preventing halitosis without causing the destruction of the normal microbial ecosystem and in inhibiting the growth of harmful microbes in the oral cavity [12]. Therefore, studies have been conducted to confirm the possibility of using probiotics for oral health, including studies that used yogurt containing probiotics [13,14] and dongchimi juice, a traditional Korean fermented food containing kimchi probiotic bacteria from pickled radish [15]. These studies showed that probiotic-containing products are comparable to conventional chlorhexidine solutions in improving oral conditions and in oral disinfection. However, in the study that used yogurt, the possibility of tooth decay due to the sugar content of the yogurt was raised. Moreover, in the study on dongchimi juice, the possibility of tooth decay due to the production of strong acid was also mentioned, and there were difficulties in maintaining a certain acidity level and in storing the juice [13].

Therefore, this study applied *Weissella cibaria* Chonnam Medical University (*W. cibaria* CMU), which is an oral probiotic isolated from the saliva of healthy children [16] in tablet form that does not contain sugar or produce strong acid. It has been shown to suppress the concentration of VSCs from halitosis-causing microorganisms through secretion of water-soluble glucan and hydrogen peroxide [17]. It has also been reported to have a significant effect on the reduction of subjective halitosis in adults with halitosis [18]. Further, since halitosis can cause serious psychosocial problems, appropriate evaluation and effective treatment are required [7]. Therefore, this study evaluated the effects of ingesting *W. cibaria* CMU on not only the reduction of subjective halitosis and improvement of oral health, but also on the improvement of psychosocial health issues such as depression, low self-esteem, and oral-health-related quality of life.

## 2. Materials and Methods

### 2.1. Subjects

The sample size required for this study was calculated using the G*Power 3.1.7 software (Heinrich-Heine-University, Düsseldorf, Germany) program. With a significance level of 0.05, a power of 0.8, and an effect size of 0.7, the minimum number of participants required for an independent *t*-test was 68, but 100 people were enrolled, considering a 30% dropout rate.

The inclusion criteria were as follows: college students over 20 years of age, with 20 or more natural teeth, and VSCs levels ≥ 1.5 ng/10 mL (concentration standard for VSCs that causes discomfort to others) [18]. The exclusion criteria included the following: subjects currently being treated for systemic diseases that may cause halitosis; diagnosed with rhinitis or sinusitis and gastritis; showing adverse reactions to lactose or milk products; regularly using probiotic products or supplements; had taken antibiotics within the last month; has dry mouth, with multiple dental caries or severe periodontal disease; has communication difficulties from hearing or vision problems; uses correction devices after orthodontic treatment; or has tongue problems. The dropout criteria included the following: subjects with compliance < 80%, those experiencing severe adverse effects, and those who took antibiotics during the study period.

### 2.2. Enrollment, Randomization, and Blinding

Participants were recruited from among college students in Gangwon province, South Korea, from 1 July to 30 September 2018, through an offline poster and a social-network service. This study was a randomized, double-blind, placebo-controlled study with a pretest-posttest design. All participants were randomly assigned to two groups using an Excel randomization program, and then randomization cards were made according to the order of subject enrollment. Each randomization card was placed in an opaque envelope that was sealed, and then the envelopes were opened according to the enrollment order, after which the participants were assigned to either the experimental group or the placebo group. A total of 100 participants were recruited, and 92 were randomly assigned to the experimental group (*n* = 49) or placebo group (*n* = 43) after excluding eight participants who refused to participate or did not meet the inclusion criteria.

The participants participated in the study without knowing whether they would be in the experimental group or the placebo group, and the researchers performed the experimental treatment in a double-blind state in which they did not know who among the participants would be in the experimental group or the placebo group. In the course of the study, the researchers recognized the participants only from their enrollment number, so the double-blind state was maintained.

During the intervention, seven participants withdrew consent due to their inability to meet the visit time, and four participants dropped out due to their use of antibiotics for wisdom-tooth extraction. Sixty-two subjects were included in the final analysis after excluding 19 participants; 11 of which had less than 80% compliance and, eight of which had inappropriate data (Figure 1).

### 2.3. Study Treatments

The oral probiotic that was tested in this study was *W. cibaria* CMU, a gray-white tablet with a unique flavor. A formulation containing 1.0 × 10^8^ colony-forming units (CFU)/g of *W. cibaria* CMU in an 800 mg tablet manufactured and provided by OraPharm Inc. (Seoul, Korea) was used. The other ingredients were isomalt, peppermint flavor, sucralose, magnesium stearate, and maltodextrin. For the placebo-group treatment, placebo tablets that did not contain *W. cibaria* CMU but had the same taste, texture, and appearance and were from the same manufacturer were used.

For a total of eight weeks, the experimental group took one tablet of the oral probiotic, and the placebo group took one tablet of the placebo, both groups once daily. The participants were instructed to put the tablet in their mouth just before going to bed in the evening and not to bite or swallow it, but to let it dissolve on their tongue for a long time, and to not eat or drink anything afterward.

Before the *W. cibaria* CMU and placebo tablets were given to the participants, they received written instructions on the following: (1) they should not ingest other commercial probiotic products; (2) they should avoid antibacterial oral rinses; and (3) they should maintain their usual brushing and eating habits. In addition, a week before the tablets were administered, the participants underwent an oral examination by a dentist at the designated dental clinic to check for oral diseases and for scaling to keep the oral environment the same. After the scaling, the participants were given the same brand of toothbrush and fluoride toothpaste for use during the study.

The *W. cibaria* CMU and placebo tablets were provided to the participants living in the dormitory by asking them to go to the dormitory cafeteria at the first-floor lobby every night between 10 p.m. and 12 p.m. Two research assistants delivered the daily dose of tablets to the participants each day according to the subject’s enrollment number. While delivering the tablets to the participants, the research assistants taught the participants how to ingest them. For the participants who did not live in the dormitory, enough tablets for two to four weeks were distributed to them in drug containers according to their enrollment number.

To improve the compliance of the participants, they were each given an intake calendar on which they were asked to record their daily intake, and were asked to take a picture of their entry on the intake calendar every two weeks for transmission to the research assistant via mobile phone. The research assistant sent each subject a short text message twice a week and conducted phone monitoring twice a week to increase their compliance. In addition, the research assistant checked the intake calendar picture sent by the subject once every two weeks.

### 2.4. Measurement and Instruments

A pre-test was performed for both the experimental group and the placebo group when the participants underwent scaling at the dental clinic a week before their ingestion of the tablets. After the participants took the oral probiotic or placebo for eight weeks, a post-test was conducted in the ninth week. The data collection was conducted by two research assistants. To increase the reliability of the measurers, they were educated on the survey method, and how to use the manual before the data collection. The data collection was conducted via a survey in a blind state, wherein they did not know whether the participants were in the experimental group or the placebo group.

Subjective halitosis was evaluated by asking the participants if they perceived halitosis. A “Yes” answer meant that the subject usually perceived halitosis, and a “No” meant that the subject did not usually perceive halitosis.

Subjective oral-health status was assessed using an instrument developed by Park [19]. The instrument consists of 10 items (overall oral health conditions, problems with chewing, gum swelling, gingival bleeding, difficulties with chewing cold or hot food, dry mouth, temporomandibular disorders, etc.), with each item rated on a five-point Likert scale, with a higher total score indicating poor oral-health status. In our study, Cronbach’s α was 0.81.

Depression was assessed using the Korean version of the Center for Epidemiologic Studies Depression Scale developed by Chon et al. [20]. The instrument consists of 20 items, with each item rated on a four-point Likert scale, with a higher total score indicating greater depression. Cronbach’s α was 0.91 in the study by Chon et al. [20] and 0.82 in our study.

Self-esteem was assessed using an instrument developed by Rosenberg [21] and translated into Korean by Jon [22]. The instrument consists of 10 items, with each item rated on a five-point Likert scale, with a higher total score indicating greater self-esteem. Cronbach’s α was 0.85 in the study by Rosenberg [21] and 0.79 in our study.

Oral-health-related quality of life was measured using the shortened version of the Oral Health Impact Profile developed by Slade and Spencer [22]. The instrument consists of 14 items, with each item rated on a five-point Likert scale, with a higher total score indicating greater oral-health-related quality of life. The Cronbach’s α was 0.97 in the study by Slade and Spencer [22] and 0.92 in our study.

### 2.5. Safety

All adverse events were monitored to assess the participants’ safety. Adverse events were also checked when the tablets were delivered daily, and when telephone monitoring was performed twice a week. When an adverse event occurred, the start time, end time, and severity of symptoms were identified, recorded, and reported to the researcher immediately.

### 2.6. Ethical Considerations

This study was conducted in accordance with the International Council for Harmonization of Technical Requirements for Pharmaceuticals for Human Use (ICH) guidelines. All subjects gave their informed consent for inclusion before they participated in the study. The study was conducted in accordance with the Declaration of Helsinki, and the protocol was approved by the Ethics Committee of Kangwon National University (IRB No.: KWNUIRB-2018-05-003-005). The participants were given a verbal and written explanation of the purpose and procedure of the study. Participants were also informed about the anonymity, confidentiality, and destruction of the collected data, as well as the possibility of subject withdrawal from the study.

### 2.7. Statistical Analysis

The data were analyzed with the following statistical methods, using SPSS version 24.0 (IBM Corp., Armonk, NY, USA). The general characteristics of the participants were analyzed by frequency and percentage, and the prior homogeneity of the experimental group and the placebo group was analyzed using the χ^2^-test or Fisher’s exact test, Mann–Whitney test, or independent *t*-test according to normality. To verify the effect of the intake of the oral probiotic on the experimental group, the χ^2^-test, Mann–Whitney test, or independent *t*-test was performed. The normality of the dependent variable was verified through the Shapiro–Wilk test.

## 3. Results

### 3.1. Homogeneity of Subjects’ General Characteristics and Dependent Variables before Treatment

The general characteristics of the participants are presented in Table 1. The verification of the prior homogeneity of the participants’ general characteristics and the dependent variable showed no statistically significant differences between the experimental group and the placebo group before the treatment (*p* > 0.05).

### 3.2. Effects on Subjective Halitosis, Depression, Self-Esteem, Oral-Health-Related Quality of Life, and Oral-Health Status

Table 2 shows the differences before and after the intervention between the experimental group and the placebo group for the study variables. In the placebo group, 50% of the participants answered “yes” when they were asked if they perceived they had halitosis after intervention, and 50% said “no.” In the experimental group, 23.5% answered “yes” to the same question and 76.5% said “no.” These figures show a statistically significant difference between the two groups (χ^2^ = 4.70, *p* < 0.030).

As for the subjective oral-health status, the difference before and after the intervention was −1.96 ± 5.50 in the placebo group and −1.82 ± 4.31 in the experimental group, which showed no significant difference between the two groups (t = −0.11, *p* = 0.911).

The difference in depression before and after the intervention was 0.92 ± 7.51 in the placebo group and 2.26 ± 8.25 in the experimental group, which showed no significant difference between the two groups (Z = −0.51, *p* = 0.605).

As for self-esteem, the difference before and after the intervention was −0.35 ± 4.66 in the placebo group and −0.79 ± 3.88 in the experimental group, which showed no significant difference between the two groups (t = 0.40, *p* = 0.688).

For oral-health-related quality of life, the difference between the two groups before and after the intervention was −2.57 ± 7.99 in the placebo group and 0.47 ± 5.01 in the experimental group, which showed a statistically significant difference between the two groups (Z = −2.09, *p* = 0.036).

### 3.3. Safety

The evaluation of the safety of the participants revealed one adverse event of mild xerostomia in the experimental group and one adverse event of mild diarrhea in the placebo group. However, after determining the start time, end time, and severity of such adverse events, they were labeled as mild symptoms and were confirmed to be unrelated to the oral probiotic used in this study.

## 4. Discussion

This study aimed to evaluate the effects of the intake of an oral probiotic on college students. This is the first study to evaluate not only the reduction of halitosis and improvement of oral health but also the psychosocial effects after the ingestion of an oral probiotic.

After the ingestion of *W. cibaria* CMU, there was a significant difference in subjective halitosis between the experimental group and the placebo group, which is consistent with previous studies [23] that showed a significant effect in the reduction of halitosis after ingestion of *W. cibaria*. These results show that *W. cibaria* is effective in reducing halitosis. Halitosis is mainly caused by VSCs in harmful microorganisms in the oral cavity [9], and the effect of the oral probiotic in inhibiting the proliferation of harmful oral microorganisms [12] appears to have reduced halitosis. The objective evaluation of halitosis through measurement of VSCs is important, but it is also important to measure subjective halitosis, as done in this study, because psychosocial problems are caused by subjectively perceived halitosis.

In this study, there was also a statistically significant difference in the oral-health-related quality of life of the participants. Many studies have been conducted on the reduction of halitosis or oral microbes after ingestion of oral probiotics, but this was the first study that measured oral-health-related quality of life, so the results cannot be compared with those of other studies. Since oral-health-related quality of life is an assessment of mental discomfort; decline of physical, mental, and social abilities; and social disadvantages related to oral health [22], it is meaningful that an oral probiotic was effective in improving the participants’ oral-health-related quality of life. In addition, higher degrees of subjective halitosis have a negative effect on oral-health-related quality of life [24]. In this study, it seems that oral-health-related quality of life improved as subjective halitosis decreased significantly in the experimental group compared to the placebo group. However, because various factors such as stress, oral symptoms, and dry mouth affect the oral-health-related quality of life of college students [24], to determine more accurately whether the decrease in the incidence of subjective halitosis in this study was induced only by the oral probiotic, it is necessary to identify and placebo the variables that influence oral-health-related quality of life.

There was no significant effect on the subjective oral-health status of the participants. Items that measure subjective oral-health status include questions about problems with chewing, gingival bleeding, and difficulties with chewing cold or hot food [19]. The subjective oral-health status of college students in this study was not poor, so the intake of the oral probiotic did not seem to have any effect on the improvement of their subjective oral-health status. However, other studies of patients with moderate to severe gingivitis have shown that oral probiotics are effective in combating oral diseases such as gingival bleeding and gingivitis [25,26]. Thus, in this study, the oral probiotic was found to have no significant effect on subjective oral-health status, but it is necessary to evaluate and confirm gum bleeding and gum status through an objective evaluation to determine oral-health status. In addition, recent systematic reviews have shown that *Porphyromonas gingivalis*, an important causative agent that causes periodontal disease and halitosis, affects important systemic diseases involving cariology, diabetology, oncology, and neurology [27]. Therefore, it is necessary to evaluate these causative bacteria together to identify oral-health status and decreased halitosis.

Halitosis also affects psychological health [5], and particularly, depression [6,7]. The results of this study show that a reduction in subjective halitosis may not immediately reduce depression. This is because among college students, depression is influenced not only by halitosis, but also by various other factors such as personality, personal vulnerability, stress, and social support [28]. These results show that although ingestion of the oral probiotic reduced halitosis, its effect on reducing depression may not appear directly. In addition, when evaluating the depression-reduction effect of the ingestion of oral probiotics according to its reduction of halitosis, it is necessary to develop oral-health-related or halitosis-related depression-measurement tools. That is, if these tools are used, the depression associated with halitosis reduction after oral-probiotic intake could be more accurately evaluated. In this study, oral-probiotic intake was used to suppress oral bacterial growth to identify the effect of depression reduction due to a decrease in halitosis. However, in addition to oral probiotics, gut microbiota have been reported to be associated with mental-health issues such as stress and depression [29,30]. The ingestion method of the oral probiotic in this study allowed it to remain in the subject’s mouth for as long as possible without swallowing it; however, this can affect not only oral microorganisms, but also intestinal microorganisms; hence, it can act as a bias in evaluating the effect of oral probiotics on depression.

In previous studies that showed that halitosis lowers self-esteem [7], it was determined that self-esteem could be improved if halitosis is reduced through oral-probiotic intake. In addition, in a study that reported that good oral-health-related quality of life increases self-esteem [31], it was inferred that if halitosis is reduced after the ingestion of oral probiotics, and oral-health-related quality of life is improved, then self-esteem can be increased. However, in this study, there was no statistically significant differences in self-esteem after ingestion of the oral probiotic. As with depression, it was found in this study that in college students, self-esteem is influenced not only by halitosis, but also by various other factors, such as locus of placebo and satisfaction with interpersonal relationships [32]. This shows that the reduction of halitosis through oral-probiotic intake in college students may not directly affect the improvement of their self-esteem.

Our study is meaningful in that it showed that ingestion of the oral probiotic was effective in reducing subjective halitosis and improving the oral-health-related quality of life of college students. It is the first study to evaluate the reduction of halitosis, oral-health status, and psychosocial effects after ingestion of an oral probiotic by college students at a life stage when companionship and intimacy are important. In addition, this study contributes to the literature by showing the possibility for the intake of oral probiotics to be used as a nursing intervention in the future by verifying its effects on reducing halitosis and improving the oral-health-related quality of life of college students. Finally, this study is significant in that the reliability of the results was improved by evaluating the effects of oral-probiotic intake through a double-blind, randomized placebo design.

However, this study had limitations in interpreting the research results. First, it is difficult to generalize the conclusions to all college students because the study was conducted only for some college students. Second, since psychological measurement tools related to the reduction of halitosis were not used, the psychological-improvement effect of the reduction of halitosis did not appear sufficiently. Finally, when measuring halitosis, we did not take objective measurements of VSCs before and after intake; hence, the objectivity of the study results could not be established.

## 5. Conclusions

The results of this study suggest that oral ingestion of *W. cibaria* CMU can help reduce subjective halitosis and improve oral-health-related quality of life, and demonstrate the possibility of oral-probiotic use by college students without any difficulties or side effects.

## Figures and Tables

**Figure 1 ijerph-18-01143-f001:**
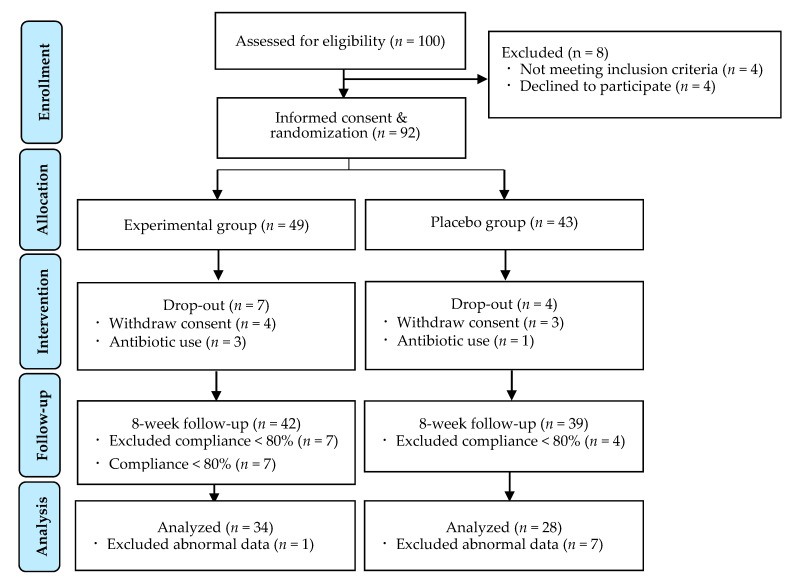
Research flow diagram.

**Table 1 ijerph-18-01143-t001:** Homogeneity of subjects’ general characteristics and dependent variables between the experimental and placebo groups before treatment (*n* = 62).

Characteristics		Placebo (*n* = 28)	Experimental (*n* = 34)	χ^2^ or Z or t	*p*
		*n* (%) or M ± SD	*n* (%) or M ± SD		
Age (year)		23.75 ± 3.42	23.44 ± 2.88	0.37	0.710
	20–25	23 (82.1)	29 (85.3)	0.73 ^†^	0.872
	26–30	4 (14.3)	3 (8.8)		
	31 and above	1 (3.6)	2 (5.9)		
Gender	Female	13 (46.4)	10 (29.4)	1.90	0.195
	Male	15 (53.6)	24 (70.6)		
Drinking	Yes	20 (71.4)	24 (70.6)	0.01	0.942
	No	8 (28.6)	10 (29.4)		
Smoking	Yes	4 (14.3)	3 (8.8)	0.45 ^†^	0.691
	No	24 (85.7)	31 (91.2)		
Brushing/day	One time	2 (7.1)	2(5.8)	2.19 ^†^	0.784
	Two	9 (32.1)	13 (38.2)		
	Three	11 (39.3)	10 (29.4)		
	Four or more	6 (21.5)	9 (26.6)		
Oral examination	Yes	11 (39.3)	14 (41.2)	0.16	0.880
	No	17 (60.7)	20 (58.8)		
Subjective halitosis	Yes	15 (53.6)	20 (58.8)	0.17	0.678
	No	13 (46.4)	14 (41.2)		
Subjective oral health status		22.57 ± 5.47	23.38 ± 6.98	−0.50	0.618
Depression		14.07 ± 8.81	14.20 ± 9.83	−0.19 ^‡^	0.848
Self-esteem		34.10 ± 6.47	33.00 ± 4.67	0.78	0.438
Oral health-related quality of life		63.39 ± 5.87	61.00 ± 6.51	−0.79 ^‡^	0.427

^†^ Fisher’s exact test; ^‡^ Mann-Whitney test; values are mean ± standard deviation or *n* (%); tested by χ^2^-test or Fisher’s exact test, Mann–Whitney test, or independent *t*-test.

**Table 2 ijerph-18-01143-t002:** Comparisons of dependent variables between the experimental and placebo groups after treatment (*n* = 62).

Variables	Group		Pre-Test	Post-Test	Difference (Post-Pre)	χ^2^ or Z or t	*p*
			*n* (%) or M ± SD	*n* (%) or M ± SD	M ± SD		
Subjective halitosis	Placebo (*n* = 28)	Yes	15 (53.6)	14 (50.0)		4.70 ^†^	0.030
		No	13 (46.4)	14 (50.0)			
	Experimental (*n* = 34)	Yes	20 (58.8)	8 (23.5)			
		No	14 (41.2)	26 (76.5)			
Subjective oral health status	Placebo (*n* = 28)		22.57 ± 5.47	20.60 ± 7.46	−1.96 ± 5.50	−0.11	0.911
	Experimental (*n* = 34)		23.38 ± 6.98	21.55 ± 5.56	−1.82 ± 4.31		
Depression	Placebo (*n* = 28)		14.07 ± 8.81	15.00 ± 8.95	0.92 ± 7.51	−0.51 ^‡^	0.605
	Experimental (*n* = 34)		14.20 ± 9.83	16.47 ± 11.07	2.26 ± 8.25		
Self-esteem	Placebo (*n* = 28)		34.10 ± 6.47	33.75 ± 5.81	−0.35 ± 4.66	0.40	0.688
	Experimental (*n* = 34)		33.00 ± 4.67	32.20 ± 6.13	−0.79 ± 3.88		
Oral health- related QOL	Placebo (*n* = 28)		63.39 ± 5.87	60.82 ± 9.40	−2.57 ± 7.99	−2.09 ^‡^	0.036
	Experimental (*n* = 34)		61.00 ± 6.51	62.47 ± 6.36	0.47 ± 5.01		

QOL: quality of life; ^†^ χ^2^-test; ^‡^ Mann-Whitney test; values are mean ± standard deviation or *n* (%); tested by χ^2^-test, Mann–Whitney test, or independent *t*-test.

## Data Availability

The data presented in this study are available on request from the corresponding author. The data are not publicly available due to privacy restrictions.

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
