# Peer review of "Effects of Oral Probiotics on Subjective Halitosis, Oral Health, and Psychosocial Health of College Students: A Randomized, Double-Blind, Placebo-Controlled Study"

_ijerph, 2021, doi:10.3390/ijerph18031143_

Round 1
Reviewer 1 Report
The introduction provides a good, generalized background of the topic that quickly gives the reader an appreciation of the wide range of applications for this technology.the manuscript and introduction should better specify the purpose of the research. Conclusions are to an extent overstated given the limitations of the study. Please revise.
I suggest to insert the following bibliography:
Porphyromonas gingivalis, Periodontal and Systemic Implications: A Systematic Review
DOI 10.3390/dj7040114
Author Response
We really appreciate your thoughtful comment for this manuscript. We have revised that in accordance with your recommendation and have highlighted it in yellow. We inserted the bibliography in accordance with your recommendation.
Reviewer 2 Report
In the Manuscript entitled “Effects of Oral Probiotics on Subjective Halitosis, Oral Health, and Psychosocial Health of College Students: A Randomized, Double-Blind, Placebo-Controlled Study” the authors investigated the effects of ingesting oral probiotics Weissella cibaria tablets on halitosis in a cohort of early adults and to examine its effects on several psycho-social indicators. The results were interesting, showing a statistically significant differences in subjective halitosis and oral health-related quality of life. The discussion is well-balanced, and the statements are supported by the data. The study fits with the aims and scope of the journal.
For greater clarity, I suggest enumerating the inclusion and exclusion criteria. Furthermore, only minor language corrections should be necessary.
Author Response
We really appreciate your thoughtful comment for this manuscript. We have revised the inclusion and exclusion criteria in accordance with your recommendation and have highlighted it in yellow
Reviewer 3 Report
I commend the authors for conducting this interesting article but I do have some minor suggestions.
Introduction:
Paragraph 3
More specifically, halitosis is mainly caused by bacterial by-products due to gram-negative anaerobic bacteria present in the tongue and their metabolites, like volatile sulfur compounds [9].
Volatile sulfur compounds are mainly composed of hydrogen sulfide (H2S), dimethyl sulfide [(CH3)2S], and methyl mercaptan…..
Please explain to readers what is dongchimi juice?
Moreover, in the study on dongchimi juice, the possibility of tooth decay due to the production of strong acid was also mentioned….
Paragraph 4
Please rephrase: It has been reported to have a significant effect on the reduction of subjective halitosis in adults with halitosis [18] by suppressing the concentration of volatile sulfur compounds from halitosis-causing microorganisms through secretion of water-soluble glucan and hydrogen peroxide, without containing sugar and producing strong acid [16].
Methods:
Why was VSC (volatile sulfur compounds) vale of ≥1.5 ng/10 mL selected?
Please rephrase: The exclusion criteria were: currently being treated for systemic diseases that may cause halitosis, diagnosed with rhinitis or sinusitis, sensitive to probiotic products or having allergies, taking antibiotics within the last one month, diagnosed with chronic gastritis, dry mouth, multiple dental caries or severe periodontal disease, installed correction devices and fixing devices after orthodontic treatment, and tongue problems.
Please rephrase: The dropout criteria were: not meeting the required 80% (44.8 times) of the number of tablet intakes provided in the experiment (a total of 56 times) and taking antibiotics during the study period.
It might be helpful to publish what the subjective oral health assessment tool is for the readers to understand the measures assessed.
Results:
Not sure about the role of religion in table 1.
The investigators have only used subjective measures. Did the investigators try to assess an objective finding like measurement of volatile sulfur compound levels?
The other bias of this study is that mental health has been associated to gut microbiome, using probiotics could help with depression not because of the halitosis effect but due to improvement of gut dysbiosis. Perhaps worth mentioning in discussion. Reference: Gut feelings: A randomised, triple-blind, placebo-controlled trial of probiotics for depressive symptoms; J Affect Disord. 2019 Jun 15;253:317-326. doi: 10.1016/j.jad.2019.04.097. Epub 2019 May 9.
Discussion:
Please rephrase: This study evaluated the effects of the intake of oral probiotics on the oral health and psychosocial health of college students for eight weeks.
In paragraph 7: Please avoid over-utilizing the phrase "this study was significant".
